# Respiratory Rehabilitation Index (R2I): Unsupervised Clustering Approach to Identify COPD Subgroups Associated with Rehabilitation Outcomes

**DOI:** 10.3390/diagnostics15162053

**Published:** 2025-08-16

**Authors:** Ester Marra, Piergiuseppe Liuzzi, Andrea Mannini, Isabella Romagnoli, Francesco Gigliotti

**Affiliations:** IRCCS Fondazione Don Carlo Gnocchi Onlus, 50143 Firenze, Italy; emarra@dongnocchi.it (E.M.); pliuzzi@dongnocchi.it (P.L.); iromagnoli@dongnocchi.it (I.R.); fgigliotti@dongnocchi.it (F.G.)

**Keywords:** chronic obstructive pulmonary disease, unsupervised clustering, pulmonary rehabilitation

## Abstract

**Background/Objectives**: Chronic obstructive pulmonary disease (COPD) is a progressive condition whose heterogeneous endotypes, clinical manifestations, and recovery pathways complicate the identification of reliable predictors of rehabilitation outcomes. Several respiratory and functional assessments are available with no consensus on the most predictive ones. While univariate markers may miss multifactorial interactions essential for prognosis, data-driven unsupervised clustering methods can integrate complex information from different sources. This study aimed to apply unsupervised clustering to identify pre-rehabilitation characteristics predictive of discharge outcomes for COPD patients undergoing pulmonary rehabilitation. **Methods**: A total of 126 COPD patients undergoing pulmonary rehabilitation were included in the analysis. Three assessments were performed at admission, namely the forced oscillation technique, spirometry, and the six-minute walk test (6MWT). The outcome was the change in 6MWT distance between admission and discharge. Unsupervised clustering methods were applied to admission variables to identify subgroups associated with outcomes. **Results**: Among the clustering algorithms tested, k-means (with N_cl_ = 2) provided the optimal solution. The resulting respiratory rehabilitation index (R2I) was significantly associated with the outcome dichotomized via the minimal clinically important difference of 30 m. Patients with R2I = 1, indicating severe functional and respiratory impairments, were associated with higher post-rehabilitation functional improvement (*p* = 0.032). While few functional parameters of 6MWT were statistically different between the groups identified by outcome, nearly all variables in the analysis exhibited significant distribution differences among the R2I clusters. **Conclusions**: These findings highlight the heterogeneity of COPD and the potential of unsupervised clustering to identify distinct patient subgroups, enabling more personalized rehabilitation strategies.

## 1. Introduction

Chronic obstructive pulmonary disease (COPD) represents a significant global health burden, leading to substantial morbidity, mortality, and health care costs [1]. Current estimates suggest a prevalence of around 3% in the general population, with projections indicating it will become the third leading cause of death and the seventh leading cause of disability-adjusted life years (DALYs) lost by 2050 [2]. In Italy, recent estimates indicate a slightly lower prevalence, affecting approximately 2–2.5% of the general population [2]. COPD is a progressive disease, often resulting in reduced quality of life and increased risk of exacerbations, hospitalizations, and mortality [3]. The current prognostic markers for COPD include a range of clinical, lung function, and imaging parameters. These may include lung capacity measures such as forced expiratory volume in the first second (FEV_1_) [4], oxygen (O_2_) saturation, and inflammatory biomarkers such as white blood cell count and C-reactive protein levels [5].

Pulmonary rehabilitation (PR) is a cornerstone intervention in COPD management, encompassing a multidisciplinary approach aimed at improving physical and psychological health. PR programs typically include exercise training, education about COPD management, and psychosocial support [6]. However, the response to PR varies widely across individuals, reflecting the significant heterogeneity of the disease. Indeed, COPD patients present with distinct phenotypes, each characterized by unique etiological, clinical, and prognostic profiles, which complicates efforts to predict rehabilitation outcomes and tailor personalized interventions [7]. Univariate markers such as FEV_1_ [8], six-minute walk test (6MWT) covered distance [9], and scales assessing symptoms like the Medical Research Council (MRC) Dyspnoea Scale [10], have been widely used to assess COPD severity and response to rehabilitation.

However, these single-dimension measures may oversimplify the multifactorial nature of COPD, failing to capture the complex interconnections between clinical, functional, physiological, and psychological factors that shape individual recovery trajectories. This complexity highlights the need for analytical approaches capable of integrating multiple data sources to provide a more comprehensive characterization of disease variability.

Supervised machine learning methods, although highly effective for prediction tasks, rely on predefined outcome labels and therefore cannot uncover latent structures or describe the variability in patient responses.

Unsupervised machine learning techniques have increasingly been used to address the challenges posed by high-dimensional, heterogeneous data [11,12,13]. Their main strength lies in identifying complex, nonlinear relationships that traditional statistical methods may overlook, enabling a more data-driven stratification of patients. Although their clinical applicability is still constrained by the need for large and high-quality datasets, limited reproducibility, and reduced interpretability, these approaches have shown promise in supporting clinical decision-making and complementing conventional assessments.

Building on this rationale, unsupervised clustering is particularly suited to explore the underlying structure of the COPD population and identify patient subgroups with distinct recovery trajectories [14,15,16]. Recent applications of unsupervised clustering in COPD have demonstrated its potential and further support its use in this clinical context [17,18]. Despite the promise of clustering approaches, challenges remain in ensuring their reproducibility and clinical applicability. Variability in patient cohorts, data sources, and clustering methodologies across studies can lead to inconsistent results, raising concerns about the robustness of identified phenotypes.

This study employs unsupervised clustering methods to stratify COPD patients from clinical information at admission in a rehabilitation unit. The resulting index is then assessed for its predictive value on rehabilitation outcomes at discharge. To address the limitations of earlier studies, a set of variables from three distinct assessment domains, i.e., the 6MWT, forced oscillation technique (FOT), and spirometry, was considered. In addition, different clustering approaches were compared to verify consistency and robustness in subgroup identification.

## 2. Materials and Methods

### 2.1. Study Design and Collection

This study was based on both a prospective observational study (conducted from 2021 to 2022) and a retrospective observational study (from 2016 to 2018) carried out at the Pulmonary Rehabilitation Unit of IRCSS Fondazione Don Gnocchi ONLUS in Florence. The studies enrolled COPD patients undergoing an outpatient pulmonary rehabilitation program (PRP). PRP was conducted in accordance with the American Thoracic Society (ATS) and the European Respiratory Society (ERS) recommendations [19] and included education, aerobic exercise training for both upper and lower limbs, and breathing retraining. The studies shared the same inclusion criteria: patients had to meet the COPD definition outlined by GOLD standards [20]; the severity of airflow obstruction ranged from moderate to very severe according to the GOLD classification; participants were former smokers in stable condition for at least four weeks prior to enrollment; and they were receiving optimal standard treatment as recommended by GOLD guidelines. Patients with recent cardiovascular events or with neuromuscular or osteoarticular diseases that limited physical exercise and/or compromised lung mechanical properties were excluded from the PRP. The studies were approved by the Research Ethics Committee (r.n.18765_oss; r.n.15217_oss). All participants provided written informed consent at the time of assessment. The variables of interest were evaluated at two time points, namely admission (T0) and discharge (T1) over 20 sessions of the pulmonary rehabilitation program.

This analysis incorporated three distinct respiratory tests (Figure 1), including the FOT [21], spirometry [22], and the 6MWT [23], each serving a specific purpose in assessing respiratory function. The FOT procedure focused on assessing respiratory impedance by recording multiple measurements while participants breathed normally. This non-invasive technique provided detailed insights into the mechanical properties of the respiratory system, helping to identify potential abnormalities or dysfunctions [21]. Spirometry provided insightful data on lung volume and air-flow dynamics [22]. The 6MWT involved participants walking briskly for six minutes while vital signs and the distance covered were recorded. This test provided insights into participants’ functional capacity and endurance, offering a practical measure of their overall cardiopulmonary health [23].

### 2.2. Data Preparation

The outcome of the study was the 6MWT covered distance variation between T0 and T1 (namely, Delta meters). Patients with missing outcome data were excluded from the analysis. In line with international clinical guidelines, specifically the ATS/ERS technical standard on field walking tests under chronic respiratory conditions [24], a minimal clinically important difference (MCID) threshold of 30 m was used to define clinically significant improvement (CSI) in the 6MWT. Consequently, the outcome was dichotomized as follows:(1)$$6MWT_{CSI}=\left\{\begin{array}{c} 0,\;\;\;\;if\;\mathrm{Delta}\;\mathrm{meters}<30\\ 1,\;\;\;\;if\;\mathrm{Delta}\;\mathrm{meters}\geq 30 \end{array}\right.$$

The independent variables of the study were collected at T0 from the three different assessment domains mentioned above, for a total of 26 variables. Specifically, within the FOT, respiratory system resistance (R_RS_) and reactance (X_RS_) were measured during inspiration at 5 Hz, along with its variation (ΔX_RS_). Moreover, inspiratory time (T_I_), the ratio of T_I_ to total time (T_I_/T_TOT_), expiratory time (T_E_), mechanical ventilation (V_E_), tidal volume (V_T_), the percentage of respiratory flow (R_F%_), and respiratory rate (R_R_) were recorded. In spirometry, functional parameters were included, such as forced expiratory volume divided by slow vital capacity (FEV/SVC), FEV_1_, total lung capacity (TLC), inspiratory capacity (IC), functional residual capacity (FRC), and residual volume (V_R_). During the 6MWT, in addition to recording the total distance walked, patients were assessed from multiple perspectives, including O_2_ levels, O_2_ saturation, the Borg Dyspnoea Scale, and the Borg Scale for limb fatigue, measured twice, before and after the test.

A preliminary analysis was adopted to discard variables showing a cross-correlation greater than 0.8. Variables with missing values were imputed using a k-nearest neighbors (kNN)-based imputer from the Scikit-learn library [25]. Then, the remaining features were standardized by removing the mean and scaling to unit variance.

### 2.3. Clustering Methods

Patients were clustered according to four different unsupervised algorithms, including k-means [26], k-medoids [27], a Gaussian mixture model [28], and BIRCH (balanced iterative reducing and clustering using hierarchies) [29]. Input data for the unsupervised models were the independent variables of the analysis.

K-means clusters data by partitioning samples in a number of groups with equal variance [26]. The algorithm was initialized with the k-means++ method (selecting initial centroids using the distribution probability-based sampling technique [30]) with the aim of minimizing the total variance contribution to the cluster. Computation was sped up using the ELKAN method (applying the triangle inequality to avoid computation of unnecessary distances [31]).

The k-medoids algorithm, a variation of k-means, partitions data into clusters by choosing representative points (medoids) and assigning each sample to the nearest medoid [27]. The algorithm was initialized with the k-medoids++ method (following an approach similar to k-means++).

The Gaussian mixture model (GMM) assumes data are generated from a mixture of Gaussian distributions [28]. It employs the expectation–maximization algorithm to estimate the distribution parameters and assigns points to clusters based on the maximum a posteriori probability [32]. The algorithm was initialized with the k-means++ method.

BIRCH constructs a feature tree with each of the nodes representing a subcluster. The feature tree expands dynamically as new data points are added [29].

For each algorithm, the number of clusters varied between 2 and 15. The number of clusters, as well as the different initializations, were compared and selected by choosing the configuration that yielded the highest silhouette score [33]. Once the optimal number of clusters was identified, the clustering algorithm was selected based on the best compromise between the silhouette score and balance in the number of patients assigned to clusters.

### 2.4. Statistical Analysis

Descriptive statistics were calculated before the imputation to provide a comprehensive overview of the effective absolute values. The median and interquartile range (IQR) values were reported for numerical variables, while for categorical variables, absolute frequencies and percentages were calculated. A comparative analysis was conducted between the subgroups identified by the dichotomized outcome. A Mann–Whitney test was performed for numerical variables, while a chi-squared test was conducted for categorical variables. After computing the cluster centroids, a second comparative analysis was conducted (Mann–Whitney test) to assess whether the outcome distributions in the cluster groups were statistically different. Later, the dichotomized outcome was compared with the cluster labels of each algorithm through a contingency table and a chi-squared analysis. Finally, on the model that reported the best results, a Mann–Whitney test was employed to evaluate whether there were statistically significant variations in the distribution of independent variables between the clusters.

## 3. Results

### 3.1. Descriptive and Univariate Results

A total of 166 patients were initially enrolled, of whom 26 were excluded due to comorbidities, resulting in 140 patients included in the study. Among these, 14 patients had missing outcome data, leading to a final sample size of 126 patients analyzed. In this final cohort (median age 77 years [IQR = 10], males: 56), 50% of participants had a *6MWT_CSI=1_* (the median value of Delta meters was 29.5 [IQR = 61]). The preliminary correlational analysis reduced the cardinality of the variables to 20. All the variables related to the FOT and spirometry did not show significantly different distributions between the two groups stratified by outcome. Conversely, among the variables of the 6MWT, O_2_ saturation and Borg Dyspnoea Scale rating were measured at the beginning, and total meters significantly differed between the groups. (Table 1).

### 3.2. Cluster Analysis

The optimization of the number of clusters conducted for each of the clustering algorithms led to identical results for all: the configuration with two clusters was the one with the highest silhouette score (Figure 2). The silhouette scores for the two-cluster configuration were 0.20 for the Gaussian mixture model, 0.14 for BIRCH, 0.12 for k-means, and 0.08 for k-medoids.

The number of patients assigned to each cluster was computed for each clustering method to assess group balancing. K-medoids and k-means clustering resulted in the most balanced distributions (N_cl0_ = 61, N_cl1_ = 65 and N_cl0_ = 60, N_cl1_ = 66, respectively); conversely, the Gaussian mixture model and BIRCH showed less cluster balance (N_cl0_ = 11, N_cl1_ = 115 and N_cl0_ = 27, N_cl1_ = 99, respectively).

Given these findings, the k-means clustering solution has been considered the most appropriate for the analysis and was referred to as the respiratory rehabilitation index (R2I).

Concerning the comparison of clustering output with the dichotomized outcome, only k-means was statistically significant (χ^2^ = 4.58, *p* = 0.032). Conversely, the continuous outcome distribution was significantly different between the two clusters (Mann–Whitney, *p* < 0.05) for all the proposed solutions. The Delta meters distribution of the two clusters resulted in a median {IQR] of 21 [46.3] and 43.5 [74], 25 [57] and 30 [60], 20 [30.5] and 30 [57], and 21 [29] and 32 [63] for the k-means, k-medoids, GMM, and BIRCH, respectively (Figure 3). A radar plot illustrating the distribution of independent variables in the two clusters has been provided exclusively for the R2I (Figure 4). Several variables significantly differed between the two identified clusters (Table 2).

## 4. Discussion

This study demonstrated that unsupervised clustering techniques can effectively stratify COPD patients into distinct subgroups based on pre-rehabilitation characteristics, offering valuable insights into rehabilitation outcomes. The outcome measure, defined as the change in 6MWT distance between admission and discharge, was dichotomized based on the MCID threshold of 30 m. The optimal clustering solution was obtained using the k-means algorithm with two clusters, resulting in the R2I. The latter, obtained from T0 data, revealed a significant association with the outcome at T1 (*p* = 0.032), showing that patients with more severe baseline functional and respiratory impairments (R2I = 1) were positively associated with a post-rehabilitation improvement in walked distance. In particular, patients in R2I = 0, compared to those in R2I = 1, presented at admission with lower overall mechanical impairment (lower respiratory resistance values and smaller variations in reactance during the test), a more favorable ventilatory pattern and lung volumes, and better functional capacity, as indicated by higher walking performance, greater exercise tolerance, and lower perceived dyspnoea. Identifying these profiles through clustering before rehabilitation could help clinicians anticipate which patients are more likely to achieve meaningful functional improvement and adapt the intensity, focus, and monitoring of PR programs accordingly, ultimately aiming to maximize individual benefits. These findings suggested that pre-rehabilitation profiling through clustering can help identify patients who are more likely to benefit from PR in terms of the 6MWT, with a significant increase. While only a few functional parameters of the 6MWT, such as total distance and O_2_ saturation, showed significant differences between the groups identified by the outcome, the R2I clusters revealed differences in nearly all pre-rehabilitation variables. These included parameters from both the FOT and spirometry, which were not evident in the outcome-based grouping, indicating that these respiratory measures play a critical role in patient stratification and may better capture the underlying heterogeneity in rehabilitation responses. Key variables that contributed most to the discrimination between R2I clusters included ΔX_RS_, FEV/SVC, and IC. This multidimensional approach goes beyond single-domain assessments used in previous studies by capturing both respiratory mechanics and functional performance, providing a more accurate characterization of patient profiles.

From a methodological perspective, this study compares clustering algorithms, including k-medoids, the GMM, and BIRCH, in addition to k-means, ultimately selected as the most appropriate solution. The use of silhouette scores to choose the optimal number of clusters ensured an objective and reproducible approach, reinforcing the validity of the identified subgroups. These methodological strengths addressed critical gaps in the literature, where clustering solutions were often hindered by inconsistent methods and insufficient validation, resulting in a lack of reproducibility and practical relevance. By applying and comparing different clustering methodologies and achieving consistent subgroup identification across algorithms, this study enhanced confidence in the robustness of the R2I for patient stratification.

The most significant practical implication of this study is the potential to personalize rehabilitation strategies for COPD patients. COPD is a highly heterogeneous condition, with patients presenting diverse clinical profiles and responses to therapy, which often limits the effectiveness of standardized rehabilitation protocols. By stratifying patients into more homogeneous subgroups based on pre-rehabilitation features, unsupervised clustering techniques can contribute to understanding the relationship between pulmonary function impairment and mechanisms of response to PR. This approach enables the design of tailored rehabilitation programs with the potential to improve rehabilitation outcomes, reduce variability in responses, and support more effective patient management in clinical practice.

## 5. Limitations

The relatively small sample size may limit the generalizability of the findings to broader COPD populations. Moreover, conducting the study in a single rehabilitation center may have introduced bias linked to the specific population characteristics or local rehabilitation protocols.

## 6. Future Directions

Future research should focus on validating the R2I across larger COPD cohorts to enhance its generalizability and clinical applicability. Further investigations could benefit from the inclusion of additional clinical and functional variables, such as psychosocial factors (e.g., anxiety and depression [34]), comorbidities (e.g., cardiac, metabolic, orthopedic, or behavioral health problems [35]), and markers of skeletal muscle dysfunction [36]. These aspects are well established as influential determinants of rehabilitation outcomes in individuals with COPD. Incorporating them into a multidimensional framework may allow for more accurate patient stratification and could enhance the overall predictive value and clinical utility of the R2I.

## 7. Conclusions

This study shows that the unsupervised clustering of multidimensional admission data enables the identification of clinically meaningful subgroups of COPD patients undergoing pulmonary rehabilitation. By integrating 6MWT, FOT, and spirometry parameters, the R2I offers a data-driven stratification tool capable of predicting rehabilitation outcomes. Specifically, patients with more severe pre-rehabilitation impairment (R2I = 0) were more likely to achieve clinically significant improvements in functional capacity, as measured by the 6MWT.

The R2I captured differences across a broad range of admission variables, many of which were not univariately associated with the outcome. These findings underscore the potential of unsupervised machine learning approaches to uncover hidden patterns in complex clinical data and support more personalized rehabilitation strategies.

## Figures and Tables

**Figure 1 diagnostics-15-02053-f001:**
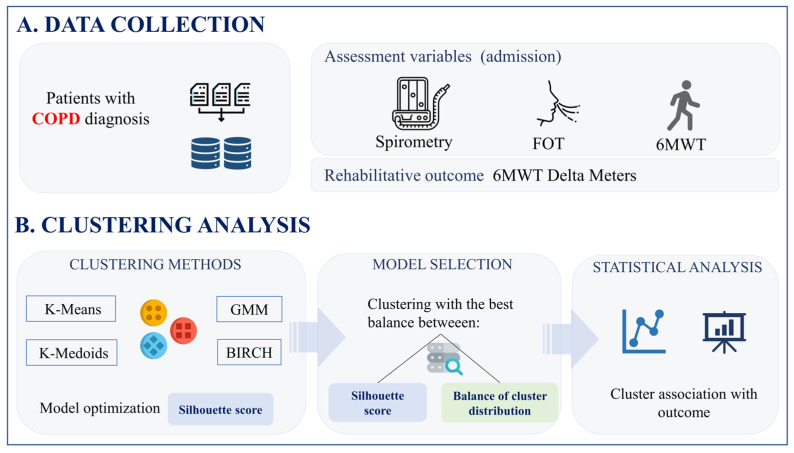
Data collection (**A**) and clustering analysis (**B**) pipeline.

**Figure 2 diagnostics-15-02053-f002:**
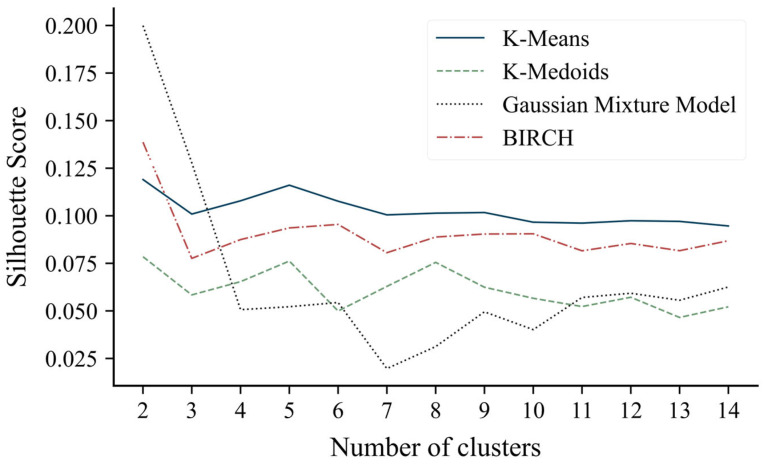
Silhouette score values as the number of clusters varies between 2 and 15.

**Figure 3 diagnostics-15-02053-f003:**
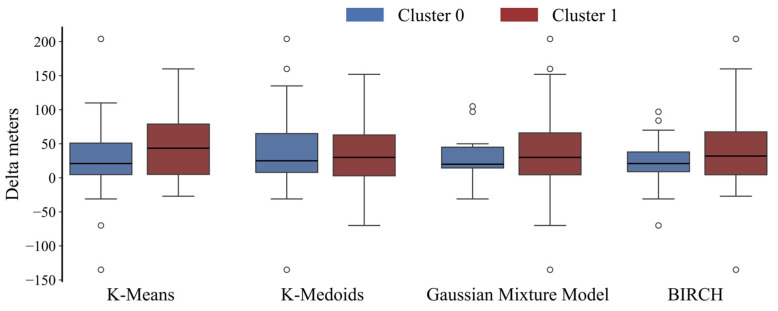
Boxplot representation of Delta meters distribution between the two clusters for each algorithm classification.

**Figure 4 diagnostics-15-02053-f004:**
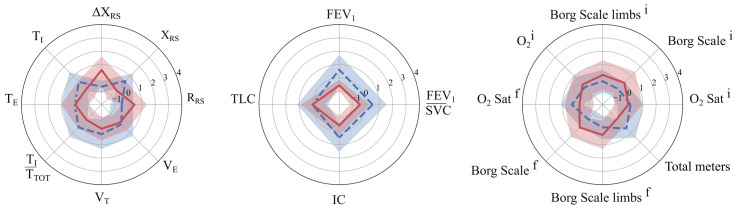
Mean and standard deviation of k-means cluster centroids. Dashed blue and solid red represent R2I equal to 0 and 1, respectively.

**Table 1 diagnostics-15-02053-t001:** Descriptive statistics of overall analysis samples: 6MWT clinically significant improvement patient sample, and no 6MWT clinically significant improvement patient sample. Comparative analysis between groups was conducted.

Variables	Total (N = 126)	6MWT_CSI=0_ (N = 63)	6MWT_CSI=1_ (N = 63)	*p*-Value
	Median [IQR] or N (%)	Median [IQR] or N (%)	Median [IQR] or N (%)	
Sex, male	1: 56 (44.4%)	1: 32 (50.7%)	1: 24 (38%)	0.151
Age, yr	77.00 [10.00]	75.00 [11.00]	78.00 [11.00]	0.373
R_RS_	4.50 [1.62]	4.37 [1.97]	4.56 [1.56]	0.238
X_RS_	−1.83 [1.33]	−1.66 [1.15]	−2.02 [1.58]	0.230
ΔX_RS_	2.13 [3.45]	1.99 [3.87]	2.51 [2.74]	0.281
T_I_	1.27 [0.53]	1.31 [0.48]	1.24 [0.55]	0.221
T_E_	2.20 [0.85]	2.31 [1.01]	2.07 [0.68]	0.135
T_I_/T_TOT_	0.37 [0.06]	0.36 [0.07]	0.38 [0.05]	0.756
V_T_	0.66 [0.33]	0.73 [0.37]	0.62 [0.30]	0.275
V_E_	11.37 [4.16]	11.27 [3.74]	11.37 [4.67]	0.560
FEV/SVC	39.50 [24.25]	40.00 [26.00]	39.00 [24.00]	0.687
FEV_1_	0.91 [0.52]	0.91 [0.52]	0.92 [0.53]	0.811
TLC	6.10 [2.68]	6.08 [3.00]	6.17 [2.54]	0.603
IC	1.66 [0.92]	1.66 [0.85]	1.63 [0.97]	0.709
O_2_ saturation ^i^	95.00 [4.00]	96.00 [3.00]	95 [3.00]	**0** **.007**
Borg Scale ^i^	0.50 [2.00]	0.00 [1.00]	0.50 [2.00]	**0** **.036**
Borg Scale limbs ^i^	0.00 [1.30]	0.00 [1.00]	0.00 [2.00]	0.491
O_2_ ^i^	0.00 [3.00]	0.00 [3.00]	0.00 [5.00]	0.709
O_2_ saturation ^f^	92.00 [7.00]	92.00 [7.00]	92.00 [7.00]	0.805
Borg Scale ^f^	5.00 [2.00]	5.00 [2.00]	5.00 [2.00]	0.877
Borg Scale limbs ^f^	3.00 [3.00]	3.00 [3.00]	3.00 [3.00]	0.877
Total meters	287.50 [148.00]	345.00 [120.00]	240.00 [112.00]	**<0.001**

Abbreviations: 6MWT, six-minute walk test; Borg Scale ^f^, Borg Dyspnoea Scale at the end of 6MWT; Borg Scale ^i^, Borg Dyspnoea Scale at the beginning of 6MWT; Borg Scale limbs ^f^, Borg Scale for limb fatigue at the end of 6MWT; Borg Scale limbs ^i^, Borg Scale for limb fatigue at the beginning of 6MWT; CSI, clinically significant improvement; FEV_1_, forced expiratory volume in the first second; FEV/SVC, forced expiratory volume divided by slow vital capacity; IC, inspiratory capacity; IQR, interquartile range; O_2_ ^i^, oxygen level at the beginning of 6MWT; O_2_ saturation ^f^, oxygen saturation at the end of 6MWT; O_2_ saturation ^i^, oxygen saturation at the beginning of 6MWT; R_RS_, respiratory system resistance; T_I_, inspiratory time; T_I_/T_TOT_, ratio of T_I_ to total time; TLC, total lung capacity; V_E_, mechanical ventilation; V_T_, tidal volume; X_RS_, respiratory system reactance; ΔX_RS_, change in respiratory system reactance. Statistically significant *p*-values are in bold.

**Table 2 diagnostics-15-02053-t002:** Comparison of median and IQR of all the variables at admission between R2I clusters. *p*-values associated with Mann–Whitney test were reported. Statistially significant values are in bold.

Variables	R2I = 0		R2I = 1		*p*-Value
	Median	IQR	Median	IQR	
R_RS_	3.74	1.44	5.06	1.61	**<0.001**
X_RS_	−1.46	0.93	−2.37	1.18	**<0.001**
ΔX_RS_	0.88	1.80	3.97	4.51	**<0.001**
T_I_	1.43	0.45	1.12	0.46	**<0.001**
T_E_	2.11	1.00	2.17	0.80	0.841
T_I_/T_TOT_	0.40	0.07	0.36	0.07	**<0.001**
V_T_	0.74	0.31	0.62	0.30	**<** **0.05**
V_E_	11.57	4.40	11.30	4.14	0.545
FEV/SVC	54.00	22.50	36.00	11.50	**<0.001**
FEV_1_	1.23	0.55	0.72	0.31	**<0.001**
TLC	5.84	3.30	6.18	2.10	0.725
IC	2.00	0.85	1.44	0.55	**<0.001**
O_2_ saturation ^i^	95.00	4.00	95.00	3.00	0.349
Borg Scale ^i^	0.00	0.50	1.00	2.00	**0.001**
Borg Scale limbs ^i^	0.00	0.00	0.00	2.00	**0.004**
O_2_ ^i^	0.00	0.50	0.00	6.00	**0.039**
O_2_ saturation ^f^	93.00	6.00	90.00	7.00	**0.003**
Borg Scale ^f^	3.00	3.00	5.00	3.00	**<0.001**
Borg Scale limbs ^f^	2.50	2.80	4.00	2.00	**0.001**
Total meters	350.00	128.00	230.00	123.00	**<0.001**

Abbreviations: 6MWT, six-minute walk test; Borg Scale ^f^, Borg Dyspnoea Scale at the end of 6MWT; Borg Scale ^i^, Borg Dyspnoea Scale at the beginning of 6MWT; Borg Scale limbs ^f^, Borg Scale for limb fatigue at the end of 6MWT; Borg Scale limbs ^i^, Borg Scale for limb fatigue at the beginning of 6MWT; CSI, clinically significant improvement; FEV_1_, forced expiratory volume in the first second; FEV/SVC, forced expiratory volume divided by slow vital capacity; IC, inspiratory capacity; IQR, interquartile range; O_2_ ^i^, oxygen level at the beginning of 6MWT; O_2_ saturation ^f^, oxygen saturation at the end of 6MWT; O_2_ saturation ^i^, oxygen saturation at the beginning of 6MWT; RRS, respiratory system resistance; TI, inspiratory time; TI/TTOT, ratio of TI to total time; TLC, total lung capacity; VE, mechanical ventilation; VT, tidal volume; XRS, respiratory system reactance; ΔXRS, change in respiratory system reactance.

## Data Availability

The data presented in this study are available upon request from the corresponding author for reproducibility purposes.

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
