# Peer review of "Respiratory Rehabilitation Index (R2I): Unsupervised Clustering Approach to Identify COPD Subgroups Associated with Rehabilitation Outcomes"

_diagnostics, 2025, doi:10.3390/diagnostics15162053_

Round 1

Reviewer 1 Report

Comments and Suggestions for Authors
  1. The manuscript mentions that several clustering algorithms were tested, with K-Means yielding the optimal result. However, it remains unclear why K-Means was ultimately preferred. Please clarify the rationale and provide supporting quantitative evidence.
  2. The rationale for adopting unsupervised clustering methods requires a more coherent and logically structured explanation. The authors should clarify why clustering is particularly suited to this study and how it contributes to addressing the heterogeneity of COPD. Strengthening this justification would help readers better appreciate the methodological choices and their added value.
  3. The figures would benefit from improved resolution, as some elements appear blurry. Additionally, the font size used for axis labels and legends is quite small, making them difficult to read. Please consider enhancing both clarity and readability for better presentation.
  4. There have been many machine learning methods applied in related fields, please discuss their advantages and disadvantages in the introduction. For example: Recognition of common non-normal walking actions based on Relief-F feature selection and relief-bagging-SVM, Isolation Forest-Voting Fusion-Multioutput: A stroke risk classification method based on the multidimensional output of abnormal sample detection, A growing model-based OCSVM for abnormal student activity detection from daily campus consumption, A Denoising Algorithm for Ultraviolet-Visible Spectrum Based on CEEMDAN and Dual-Tree Complex Wavelet Transform.
  5. The use of a 30-meter threshold in the 6MWT to define meaningful improvement is appropriate based on prior literature, but the manuscript should explicitly cite supporting clinical guidelines or studies to justify why this threshold is suitable.
  6. The current experimental evidence is sparse. Including more extensive experiments—such as comparisons with more clustering methods, ablation studies would provide a more comprehensive assessment of the proposed approach.

Reviewer 2 Report

Comments and Suggestions for Authors

Review report

I have read the article entitled "Respiratory Rehabilitation Index (R2I): Unsupervised Clustering Approach to Identify COPD Subgroups Associated with Rehabilitation Outcomes" with great interest. The manuscript addresses the important and timely topic of patient stratification in COPD rehabilitation using machine learning methods, specifically unsupervised clustering, to develop a data-driven index (R2I) that may predict rehabilitation outcomes. The study has the potential to advance personalized medicine in pulmonary rehabilitation. I have some suggestions for improvement:

  1. In the introduction, I recommend including more detailed and up-to-date epidemiological data on the incidence and prevalence of COPD in Italy. While the current introduction provides global estimates and projections, it would strengthen the manuscript to contextualize the study population with national data.
  2. In the Materials and Methods section, you state that “Patients with comorbidities that could limit participation in the PRP were excluded.” For greater clarity and reproducibility, please specify which comorbidities were considered exclusion criteria. Additionally, I recommend including a flowchart summarizing the inclusion and exclusion process, as well as the number of patients at each stage. Such a flowchart would enhance the transparency of patient selection and strengthen the methodological rigor of the manuscript.
  3. I recommend that the authors consider separating the “Limitations” and “Future Directions” into distinct sections. This will improve the clarity and readability of the manuscript, allowing readers to clearly distinguish between the constraints of the current study and the recommendations or perspectives for future research.
  4. For the Future Directions section, please consider adding that future studies may include psychosocial factors, comorbidities, and markers of skeletal muscle dysfunction, as these variables are also known to significantly influence rehabilitation outcomes in COPD. Including such multidimensional data could further enhance patient stratification and improve the predictive value of the Respiratory Rehabilitation Index.

Round 2

Reviewer 1 Report

Comments and Suggestions for Authors

none

Reviewer 2 Report

Comments and Suggestions for Authors

 Accept in present form